# Multiomics of *GCN4*-Dependent Replicative Lifespan Extension Models Reveals Gcn4 as a Regulator of Protein Turnover in Yeast

**DOI:** 10.3390/ijms242216163

**Published:** 2023-11-10

**Authors:** Blaise L. Mariner, Daniel P. Felker, Ryla J. Cantergiani, Jack Peterson, Mark A. McCormick

**Affiliations:** 1Department of Biochemistry and Molecular Biology, School of Medicine, University of New Mexico Health Sciences Center, Albuquerque, NM 87131, USAdanfelker@gmail.com (D.P.F.); ryla.cantergiani@nyulangone.org (R.J.C.);; 2Department of Chemical and Biological Engineering, University of New Mexico, Albuquerque, NM 87131, USA; 3Autophagy, Inflammation, and Metabolism Center of Biomedical Research Excellence, University of New Mexico, Albuquerque, NM 87131, USA

**Keywords:** Gcn4, ATF-4, ATF4, tRNA synthetase, autophagy, ubiquitin–proteasome system

## Abstract

We have shown that multiple tRNA synthetase inhibitors can increase lifespan in both the nematode *C. elegans* and the budding yeast *S. cerevisiae* by acting through the conserved transcription factor Gcn4 (yeast)/ATF-4 (worms). To further understand the biology downstream from this conserved transcription factor in the yeast model system, we looked at two different yeast models known to have upregulated Gcn4 and *GCN4*-dependent increased replicative lifespan. These two models were *rpl31aΔ* yeast and yeast treated with the tRNA synthetase inhibitor borrelidin. We used both proteomic and RNAseq analysis of a block experimental design that included both of these models to identify *GCN4*-dependent changes in these two long-lived strains of yeast. Proteomic analysis of these yeast indicate that the long-lived yeast have increased abundances of proteins involved in amino acid biosynthesis. The RNAseq of these same yeast uncovered further regulation of protein degradation, identifying the differential expression of genes associated with autophagy and the ubiquitin–proteasome system (UPS). The data presented here further underscore the important role that *GCN4* plays in the maintenance of protein homeostasis, which itself is an important hallmark of aging. In particular, the changes in autophagy and UPS-related gene expression that we have observed could also have wide-ranging implications for the understanding and treatment of diseases of aging that are associated with protein aggregation.

## 1. Introduction

Many organisms show a decline in their ability to maintain protein homeostasis as they age—a hallmark of aging [1,2]. Many genes and processes influence protein synthesis and protein degradation, both of which are integral for maintaining protein homeostasis [3]. Simple model organisms have repeatedly been found to have increased lifespans under conditions that inhibit protein synthesis [4,5]. In yeast and worms, reducing translation with RNAi or the deletion of ribosomal proteins can lower translation and extend lifespan [4,5,6].

Increased protein degradation, the other leg of protein turnover, has repeatedly been found to increase lifespan in simple model organisms [7,8,9]. One process involved in protein degradation, the ubiquitin–proteasome system (UPS), degrades many proteins in eukaryotic organisms and plays a specific role in degrading dysfunctional, damaged, and misfolded proteins [7,8,10]. Elevated activity of the UPS system has been shown to increase lifespan in multiple model organisms [5,7,11,12,13,14,15]. Another process of protein degradation, namely autophagy, is a bulk-degradation process that degrades cellular components and organelles [16]. Although macroautophagy is often considered in the context of bulk degradation of cellular components, other types of autophagy can play more specific and nuanced roles [9,17,18,19,20,21,22,23,24]. Macroautophagy (hereafter ‘autophagy’) is the bulk degradation process that has been studied the most to date in the context of aging [9]. Specific autophagic genes and their roles in aging and disease phenotypes have recently been reviewed in more detail [25]. Besides aging, increasing autophagy and the UPS have been shown to improve health in animal models with protein aggregate-type diseases [7,26,27,28,29,30].

The transcription factor Gcn4 (general control non-de-repressible kinase 4) in the budding yeast *S. cerevisiae* regulates gene expression in response to amino acid stress, and its functional ortholog, namely activating transcription factor 4, responds to various stressors in the nematode *C. elegans* (ATF-4), as well as in mammals (ATF4) [31,32,33]. Gcn4/ATF-4 is necessary for increased lifespan in multiple contexts, including treatment with tRNA synthetase inhibitors in both yeast and worms [34]. Gcn4 is also necessary for increased lifespan in yeast strains lacking ribosomal genes, such as *RPL31A* and *RPL20B* (Ribosomal 60S subunit protein large) [6,34,35,36]. The discovery that some ribosomal protein deletions increase the replicative lifespans of yeast through *GCN4* first demonstrated *GCN4*’s role as a gerontological gene [5,6]. Since then, the upregulation of Gcn4/ATF-4 in yeast and worms has been shown to be necessary or sufficient for lifespan extension in several different contexts [34,35,37]. Further, several types of long-lived mice have been shown to have elevated levels of ATF4 relative to those of normal-lived mice [38,39]. The way in which this nutrient-responsive transcription factor extends lifespan has yet to be defined, although our recent work has shown that elevated ATF4 levels increase protein degradation mechanisms in mammalian cells [40].

Gcn4 translation is regulated via four upstream open reading frames (uORFS) in yeast, and this mechanism of translational regulation is conserved through humans, with uORF regulation leading to similarity between worm and mammalian orthologs (Figure 1A) [33,35,41,42]. Reduced translation initiation can lead to the skipping of these regulatory uORFs and the increased translation of Gcn4/ATF-4/ATF4 [31,32,33,40,43,44,45,46].

Our work shows that tRNA synthetase inhibitors can upregulate Gcn4/ATF-4 and increase lifespan in yeast and worms [34]. This increase in Gcn4 translation upon tRNA synthetase inhibitor treatment depends on *GCN2* (General control non-de-repressible kinase 2) (Figure 1B) [34]. When Gcn2 senses uncharged tRNA, it phosphorylates eukaryotic initiation factor 2 α (eIF2α), inhibiting translation initiation [36,40,47,48]. While this process lowers most overall translation, due to the uORF regulation of Gcn4 and its orthologs, it leads to the increased translation of Gcn4/ATF-4/ATF4 [31,32,33,40,44].

Given that we now have multiple ways of increasing yeast replicative lifespan through *GCN4*, we sought to uncover the effects Gcn4 on differential expression in two different long-lived yeast models to try to identify shared signatures of *GCN4*-dependent increased lifespan.

## 2. Results

### 2.1. GCN4 Impacts Proteins Related to Amino Acid Biosynthesis in Yeast

Four uORFs in the *GCN4* 5′ untranslated region regulate *GCN4* translation initiation [32,33,45,46]. Yeast treated with borrelidin, a threonyl tRNA synthetase inhibitor, and yeast deleted for *RPL31A*, which encodes a protein of the large ribosomal subunit, both have increased Gcn4 translation and increased replicative lifespan [5,6,34,49,50]. We designed a proteomic experiment analyzing the *GCN4*-dependent changes in the yeast proteome, featuring a block-design and considering drug treatment, the *RPL31A* genotype, and the *GCN4* genotype (Figure 2A) [51]. In total, we analyzed 27 samples across these 6 conditions, resulting in 4–5 replicates per condition (Appendix A). Simple volcano plots showing differential protein abundance between the conditions of interest can be seen in Appendix A. Based on those analyses, it can be seen that borrelidin-treated yeast only had seven proteins with significantly differential abundance in comparison to either of its two normal-lived counterparts (specifically wild type + vehicle and gcn4Δ + borrelidin) (padj < 0.05) (p_adj_ < 0.05) (Appendix A). However, a linear model fitted to Gcn4 levels (Design = ~Gcn4 levels + Condition) revealed 44 proteins differentially regulated by Gcn4 (p_adj_ < 0.05) (Figure 2B).

Gene ontology analysis of proteins whose abundances were linked to the presence of Gcn4 included many biological processes related to cellular amino acid biosynthesis and protein folding (FDR < 1) (Appendix A), as summarized in an edge-node graph of over-represented biological process gene ontology categories (Figure 2C) [52,53]. These data agree with the widely known regulation of amino acid biosynthesis by Gcn4 [31,32,33,54]. The most significantly differentially expressed protein found was Cpa2, which encodes carbamyl phosphate synthetase A, an enzyme involved in the synthesis of the arginine precursor citrulline [55,56]. Other amino acid biosynthesis proteins, namely Aat2, Arg4, Arg7, Asn1, Asn2, His1, His4, His5, Hom2, Hom3, Idp1, Leu4, Lys12, Lys20, Leu9, Trp2, Trp3, Trp4, and Trp5, were upregulated in the long-lived yeast strains (Figure 2D) [57]. In our dataset, we found that Hsp78, which prevents misfolded protein aggregation, and Sgt2, which interacts with prion aggregates and is an amyloid sensor, also showed differential protein abundances in the long-lived yeast strains [58,59,60,61,62,63,64,65]. Altogether, these data suggest that *GCN4* impacts the transcription of genes involved in protein synthesis and stabilization, which is in agreement with the results of other studies [34,66,67,68].

### 2.2. GCN4 Impacts the Transcription of Protein Degradation Mechanisms in Yeast

Next, we sought to increase the coverage of our proteomics experiment and apply the same block design to an RNAseq experiment. Here, we used 96 samples across the same 6 conditions (Figure 2A). Principal component analysis of the samples showed the clustering of samples receiving the same treatment (Figure 3A). Principal component 1 was seemingly able to capture the variances between the different genotypes, while principal component 2 captured the variance in the transcriptomes impacted by translation. Although principal component 2 was able to cleanly capture the variance between the *rpl31aΔ* and *gcn4Δ rpl31aΔ* yeast, it was unable to cleanly capture the differences between vehicle- and borrelidin-treated yeast using either the wild-type or *gcn4Δ* yeast. Single-condition comparisons of particular interest for this RNA sequencing study can be seen in volcano plots in Appendix A. We found that 12 genes were consistently differentially expressed between the two long-lived yeast strains and their *GCN4*-deleted normal-lived counterparts (Appendix A). We also found many genes to be consistently differentially expressed between *RPL31A*-deleted yeast and its two normal-lived counterparts (wild type and *gcn4Δ rpl31aΔ*) (Appendix A).

Using a linear model fitted to Gcn4 levels, identical to that used in the proteomics experiment, we were able to uncover 304 genes either up- or downregulated under changed Gcn4 levels (p_adj_ < 0.01, Figure 2B). *YFL067W*, the most significantly differentially expressed gene, encodes a protein of unknown function and is not known to be conserved to other organisms [69]. *YSC83*, the second most significantly differentially expressed gene, is a non-essential mitochondrial protein of unknown function, although it is known to change expression during meiosis and located proximally to *ARG4* in the yeast genome [70]. *ALD6* is significantly downregulated in yeast with high Gcn4 translation and known to negatively regulate autophagy and impact vacuolar acidity [71,72]. *PRB1* and *TPS2* are also downregulated in yeast with increased Gcn4 translation, and these genes are known to be involved with the stress response, protein aggregation, proteolysis, and autophagy [73,74,75,76,77]. Although these genes are known to have positive and negative influences on autophagic flux in different contexts, it is clear that Gcn4 is impacting the differential expression of genes relating to autophagy. The heat-shock protein chaperones *HSP26*, *HSP42*, and *HSP12* are downregulated in the yeast strains with higher Gcn4 translation (Figure 3B) [78]. This further indicates the change in how the yeast strain is handling proteins after their translation.

The genes upregulated (FDR < 1) in response to increased Gcn4 translation were then plotted on an edge-node graph using GeneMania (Figure 3C) [52,53]. The proteasomal core complex and process utilizing autophagic mechanism were two of the most highly over-represented biological process ontology categories in this analysis. Among these, *MEH1* is known to stimulate microautophagy and positively impact vacuolar acidification [79]. *COG1* is known to impact protein trafficking through the fusion of transport vesicles [80,81,82]. *PRE6*, a highly conserved gene found here to be greatly increased with high Gcn4 translation, is a structural component of the yeast 20S proteasome [83]. Notably, *PRE6* transcripts were upregulated in the borrelidin-treated wild-type yeast in comparison to either vehicle-treated wild-type yeast or treated yeast without *GCN4*. Similarly, *rpl31aΔ* yeast had increased *PRE6* mRNA levels in comparison to *gcn4Δ rpl31aΔ* yeast, as is the case with a number of genes depicted in the heatmap provided (Figure 3D).

Overall, there were nine genes with significantly different abundances of both mRNA and protein from their respective linear models fitted to Gcn4 translation (Appendix A). *ARO4* (AROmatic amino acid requiring 4), *ILV6* (IsoLeucine Valine 6), *HIS4* (HIStidine requiring), *HOM3* (HOMoserine requiring), *IDP1* (Isocitrate dehydrogenase NADP-specific), *LYS20* (LYSine requiring), and *HOM2* (HOMoserine requiring) were the genes with both increased transcript levels (p_adj_ < 0.05) and increased protein abundance (p_adj_ < 0.05) in these studies, further confirming the known role of Gcn4 in regulating amino acid biosynthesis. The other two genes, namely *HSP78* (Heat Shock Protein) and *HIS1* (HIStidine), had significantly decreased mRNA in the RNAseq and increased protein abundance in the proteomics from their linear model fits, respectively.

### 2.3. Conserved Genes Involved in Protein Degradation Are Differentially Expressed in This Yeast Dataset

We recently found that proteasomal activity and autophagy are upregulated in response to borrelidin treatment in mouse embryonic fibroblasts (MEFs) [40]. Wipi2 was among many autophagy genes recently shown to be strongly upregulated with increased ATF4 translation in MEFs, and its yeast ortholog (*ATG21*) is shown to be greatly upregulated in this study (p_adj_ = 0.007). This gene is known to be important in the induction of autophagy, and its differential expression in both of these datasets is consistent with the possibility that autophagy may be regulated similarly downstream of *GCN4* in yeast and *Atf4* in mammals [84,85,86,87,88,89].

Several yeast genes related to proteasomal degradation, including *RAD28*, *HSE1*, *VPS27*, *DFM1*, and *RMD5*, are differentially expressed in this yeast dataset with increased Gcn4 translation (p_adj_ < 0.01), and they also have orthologs that have previously been shown to be differentially expressed in MEFs with increased Atf4 translation (p_adj_ < 1 × 10^−10^). *RAD28* deleted yeast appear to have increased mutations upon UV treatment [90]. *HSE1* and *VPS27* form a complex and sort ubiquitinated membrane proteins for degradation [91,92]. *DFM1* is a required component of the ER-associated protein degradation pathway that removes aggregated proteins [93,94]. *RMD5* is a glucose-induced degradation deficient protein, a part of an E3 ubiquitin ligase [95,96]. Altogether, these data suggest that the protein degradation regulation downstream of increased *GCN4*/*Atf4* may show conserved patterns of gene regulation in yeast and mammals.

## 3. Discussion

In order to further understand the shared mechanisms of *GCN4*-dependent delayed aging, we measured both proteomic and transcriptomic changes correlated with increased Gcn4 translation in parallel with two different Gcn4-dependent long-lived yeast models. Gcn4 and its orthologs are known to impact protein translation and amino acid biosynthesis, so it is unsurprising that we observed differential abundances of proteins related to these processes [34,40,66]. Our novel finding of significant changes in Hsp78 and Spt2 expression suggest that these long-lived yeast may have altered responses to protein aggregation.

Aligning with the published findings that autophagy is necessary for extended lifespan in a *GCN4*-over-expressing yeast strain and elevated ATF4 increases autophagy in mammalian cell cultures, it is not surprising that we found the differential expression of genes known to impact autophagy [37,40,97]. Our datasets identify potential genetic activators of autophagy under conditions of increased Gcn4 translation, especially *ATG21*, *COG1*, and *MEH1*. Elevated UPS activity has been associated with increased lifespan, and genes involved in proteasomal activity have been shown to change significantly via the upregulation of Gcn4’s functional mammalian ortholog, namely ATF4, so further inquiry into how the UPS changes downstream of Gcn4 is warranted [5,7,8,12,13,14,28,40].

Our transcriptomic analysis uncovered differentially expressed genes involved in amino acid biosynthesis, autophagy, the structure of the proteasome, protein folding, and protein aggregation, likely as a response to the sensed depletion of amino acids and needed protein maintenance [40,98]. Notably, a handful of genes in the yeast ubiquitin-dependent protein catabolic biological process ontology category were differentially expressed in both this dataset and a published borrelidin-treated mammalian cell RNAseq dataset [40]. *ATG21* and its mammalian ortholog Wipi2 (which is a gene involved in the induction of autophagy) are also upregulated with increased Gcn4 and ATF4 translation in both the yeast RNAseq dataset shown here and the published borrelidin mammalian cell dataset. These data suggest that both yeast Gcn4 and mammalian ATF4 genetically impact the transcription of genes involved in both the UPS and autophagy, two major protein degradation processes.

Further studies exploring the impacts of these *GCN4* targets on protein homeostasis in the context of aging are warranted. Although some genes and proteins with differential abundance in these long-lived yeast strains were identified, we still do not know which processes downstream of Gcn4 are responsible for the increased lifespan. Our interpretation of these data assumes that borrelidin, as well as the long-lived phenotype associated with it, are associated with its ability to upregulate Gcn4 translation and downstream transcriptional activity, in part due to the *GCN4* dependence of increased lifespan in borrelidin-treated yeast. Although these global studies were conducted with high sample numbers for each condition in both our proteomics and RNA sequencing studies, further studies confirming individual gene and protein hits of particular interest using standard methods, including qPCR or Western blot, are suggested.

Overall, these data indicate that drugs which increase Gcn4 and have been shown elsewhere to increase its orthologs could be used to improve protein turnover. This suggests the possibility that these drugs have potential to treat diseases characterized by protein aggregation, such as Huntington’s disease, Parkinson’s disease, and Alzheimer’s disease.

## 4. Materials and Methods

### 4.1. Yeast Culture

Yeast strains were pulled from the YKO deletion collection and grown at 30 °C in sterile YPD medium (1% yeast extract (BD Biosciences, San Jose, CA, USA, Cat No. 212730), 2% peptone (ThermoFisher Scientific, Detroit, MI, USA, Cat No. 211820), 2% dextrose (VWR, Solon, OH, USA, Cat. No. 97061-172)) via orbital shaking at 250 RPM with flasks of sterile water to maintain humidity for all protein and RNA extractions, as outlined below.

### 4.2. Protein Extraction, Mass Spectrometry, and Proteome Analysis

Yeast protein was extracted from yeast found to have an OD600 of between 0.3 and 0.9, with further guidelines taken from the Clontech Yeast Protocols Handbook [99]. These yeast should, thus, have a replicative age of ~1 generation, with the majority having divided zero or one times. The concentrations of proteins were confirmed via the Bradford assay using the Coomassie Plus Kit (Thermo Scientific, Detroit, MI, USA, Cat. No. 23236). Protein samples were sent to the University of Arkansas for Medical Science’s (UAMS) Proteomics Core Facility as part of an IDEA National Resource for Quantitative Proteomics Nationwide Voucher Program Award. The proteomics analysis of mass spectrometry data from the 27 samples that passed quality control involved BiocManager’s DEP package (Appendix A) [100]. 

### 4.3. RNA Extraction, Sequencing, and Analysis

RNA was extracted using the Yeast RNA miniprep kit acquired from Zymo Research (Cat# R1002) in accordance with the manufacturer’s protocols during log-phase growth between an OD600 nm of 0.6–0.8, as measured via a Victor NIVO [101]. DMSO was used as the vehicle in all cases for drug-treated yeast. In total, 5 μM and 10 μM measurements of borrelidin were used in the borrelidin-treated samples, in accordance with the upregulation of Gcn4, as previously published [34]. Transcriptomes were sequenced with the Illumina NextSeq 500 at the Center for Evolutionary and Theoretical Immunology. We first used fastp for quality control and the removal of forward and reverse adapters [102]. After that step, we used HISAT2 and featurecounts to align and quantify these trimmed files [103,104]. Data analysis and differential expression utilized R programming and associated libraries, such as limma and DESEQ2 [100,105,106,107,108].

## Figures and Tables

**Figure 1 ijms-24-16163-f001:**
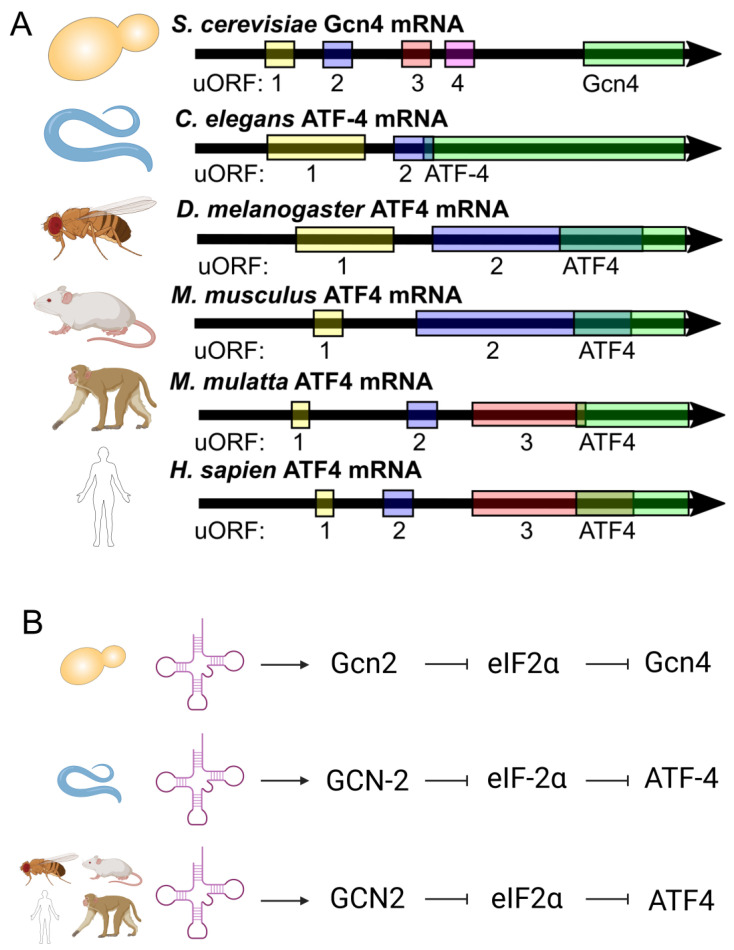
Gcn4/ATF-4/ATF4 and its highly conserved nature of activation. (**A**) Gcn4/ATF-4/ATF4 is regulated via upstream open reading frames (uORFs). (**B**) The uncharged tRNA sensor, namely Gcn2/GCN-2/GCN2, can lead to a signal cascade and the translation of Gcn4/ATF-4/ATF4 in a highly conserved manner.

**Figure 2 ijms-24-16163-f002:**
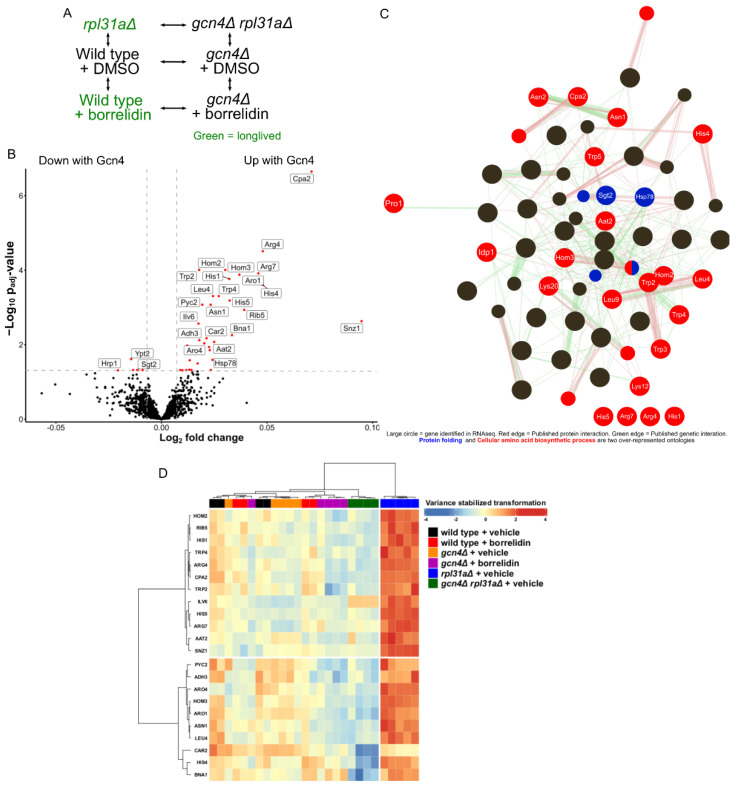
Proteomic analysis of long-lived yeast. (**A**) Block design used for the genomic analysis. (**B**) Volcano plot showing the results of the linear model fitted to Gcn4 translation. Red dots indicate that the gene is significantly differentially expressed from the linear model fitted to reported Gcn4 translation levels (p_adj_ < 0.05. Log_2_ fold change > |0.01|). (**C**) GeneMania interaction and biological process analysis of differentially abundant proteins (p_adj_ < 0.01). (**D**) Heatmap of the differentially abundant proteins (p_adj_ < 0.01) from the linear model fitted to Gcn4 translation.

**Figure 3 ijms-24-16163-f003:**
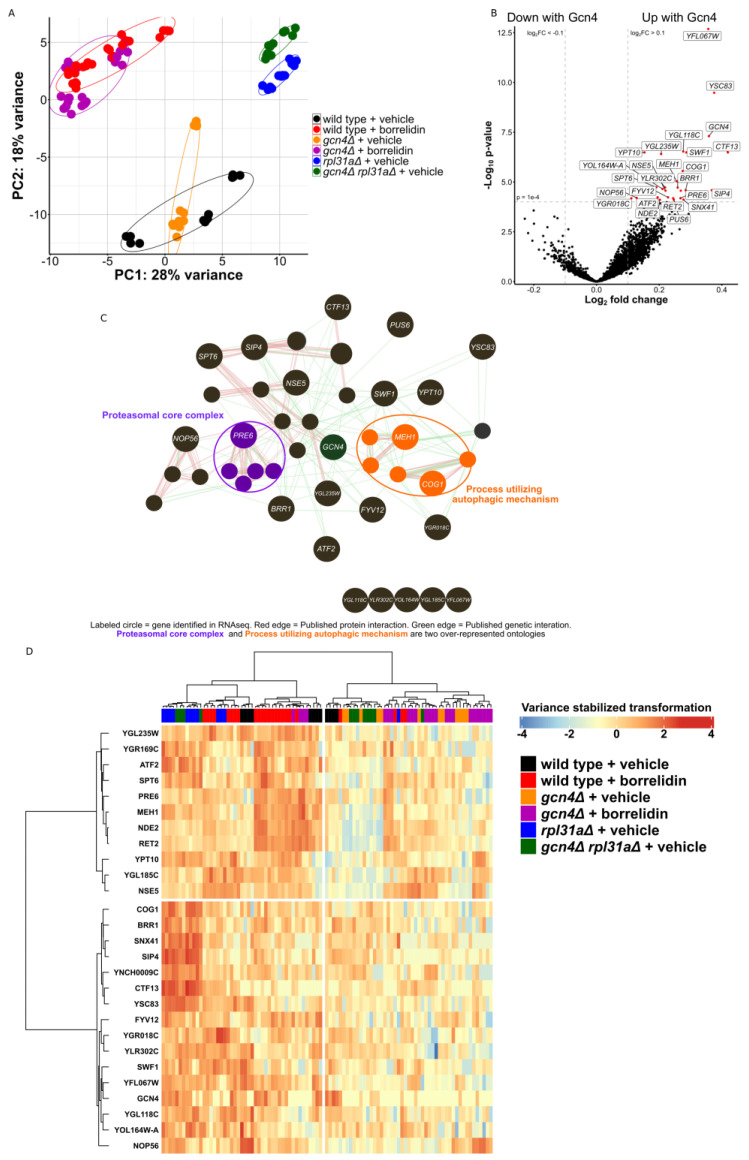
RNA sequencing of long-lived yeast. (**A**) Principal component (PC) analysis of the sequenced and analyzed samples. (**B**) Volcano plot showing the result of the linear model fitted to Gcn4 translation. Red dots indicate that the gene is significantly differentially expressed from the linear model fitted to reported Gcn4 translation levels (p_adj_ < 1 × 10^−4^, Log_2_ fold change > |0.1|). (**C**) GeneMania interaction and biological process analysis of these differentially expressed genes (p_adj_ < 1 × 10^−4^). (**D**) Heatmap showing the mRNA abundance of the 27 genes found to be differentially expressed from the linear model fitted to Gcn4 translation (p_adj_ < 1 × 10^−4^).

## Data Availability

All genomic data described in this study are available in the Appendix A or the Gene Expression Omnibus (GEO) with accession number #GSE242739.

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
