# Peer review of "Multiomics of GCN4-Dependent Replicative Lifespan Extension Models Reveals Gcn4 as a Regulator of Protein Turnover in Yeast"

_ijms, 2023, doi:10.3390/ijms242216163_

Round 1

Reviewer 1 Report

Comments and Suggestions for Authors

Mariner et al. investigated GCN4 dependent replicative lifespan expansion models using proteomic and transcriptome approaches. GCN4 is a critical transcription factor involved in responding to various stresses and plays a role in diverse biological pathways, including autophagy, chaperones and apoptosis. The authors utilized yeast models in which gcn4 was deleted, combined with other methods of translation inhibition, to identify potential protein and RNA targets and/or pathways altered by GCN4 translation. Their research idea is intriguing and holds promise for advancing our understanding of aging and stress responses. However, there is a need for substantial improvement in data presentation and the design of several major experiments. Here are the major and minor points for enhancing their manuscript.

Major points

-         The authors should provide specific details about the conditions of the yeast used in their study, including information about the passage and age of the yeast. The conditions they used (e.g. gcn4 deletion) impacts on replicative lifespan, it is important to differentiate between early (young) and late (many passages) replicated cells, as they may have different RNA and protein expression profiles.

-         In the both proteomic and RNA-seq volcano plots, they should verify the units on the x axis (0.05 and 0.2) and ensure that they are correct.  Also, they should include cutoff lines for both x and y axes. Additionally, it appears that the cutoff values for p-values are different between the two plots (2B and 3B). This should be clarified.

-         The authors presented only one volcano plot (B) in Fig 2 and 3 despite listing six samples. They should consider conducting further analysis to monitor the changes in each sample and the combined effect of rpl31a, tRNA synthase and gcn4 deletion. It would be valuable to show correlations between the samples, particularly as they applied single and double stress conditions (e.g. rpl31a deletion vs gcn4 + rpl31a double deletion).

-         The authors concluded that ‘ GCN4 impacts the transcription of genes involved with protein synthesis and stabilization….’  Using proteomic analysis alone is not sufficient to conclude that GCN4 directly impacts transcription. The altered protein and RNA levels could be the results of indirect or secondary effects of GCN4 as well.

-         The authors should include validations for their designed experiments. For example, in the case of proteomics, they should validate the altered protein levels by WB (e.g. hsp78). Additionally, they should provide WB images to confirm the rpl31a and gcn4 level in supplementary figures.

-         For both proteomics and RNA-seq, I recommend including heatmap of DEG from each experiment to assess the consistency of results

-         For RNA-seq analysis, they should also include validation of their results (e.g. qPCR)

-         They listed PRE6 based on TKM. It would be helpful to show additional genome browser views (e.g. using UCSC or igv browser) to visualize the differences. Also, qPCR validation needs to be done.

Minor points

-         They should further explore the correlation between RNA-seq and proteomic analysis. Even a simple Venn diagram could help clarify the relationship between the two datasets

-         Line 163-164, typo Figure 3D -> 3C.

-         The manuscript did not state Figure 3B in the text (it should be in line 160-161).

Reviewer 2 Report

Comments and Suggestions for Authors

The authors begin by discussing the connection between inhibition of protein synthesis or elevated protein degradation and lifetime extension. They describe the key role of Gcn4/ATF-4/ATF4 and point out that upregulating this protein results in lifetime extension. The authors discuss how upstream open reading frames regulate GCN4 translation. Inhibiting tRNA synthetase upregulates Gcn4/ATF-4 and increases lifespan. 

Treatment with borrelidin, a threonyl tRNA synthetase inhibitor and deletion of RPL31A increase Gcn4 translation. Proteomic analysis identified proteins that were differentially regulated by Gcn4 and these proteins were enriched for proteins involved in amino acid biosynthesis and protein folding. An RNAseq experiment and principal component analysis were carried out. 304 genes were either upregulated or downregulated in response to increased Gcn4 levels. These genes include genes with roles in protein aggregation, proteolysis and autophagy and three protein chaperones. Genes with roles in amino acid biosynthesis were upregulated at both the mRNA and protein levels. 

The authors suggest that drugs that increase Gcn4 levels might be used to treat protein aggregation diseases.

The article is well written and the findings clearly support the conclusions. Due to the conservation of Gcn4/ATF-4/ATF4 the findings are of interest to a wide readership. I would recommend publication of the article. There are a few typo-type errors.

159: chaperons should be chaperones

223: fo should be of

Figure 1: part of the legend has been displaced 

Round 2

Reviewer 1 Report

Comments and Suggestions for Authors

I appreciate that the authors corrected and clarified my comments, and added new analysis. I strongly recommend that the authors carefully read through and smoothen the manuscript before final submission (e.g. I see multiple identical sentences stating, "Single-condition comparison volcano plots can be seen in Supplemental Figure #"). 
